# *Impressions*: Understanding Visual Semiotics and Aesthetic Impact

**Julia Kruk** 🐝    **Caleb Ziems** 🌲    **Diyi Yang** 🌲

🐝 Georgia Institute of Technology, 🌲 Stanford University

jkruk3@gatech.edu, {cziems, diyiy}@stanford.edu

## Abstract

Is *aesthetic impact* different from beauty? Is visual salience a reflection of its capacity for effective communication? We present **Impressions**,[1] a novel dataset through which to investigate the semiotics of images, and how specific visual features and design choices can elicit specific emotions, thoughts and beliefs. We posit that the impactfulness of an image extends beyond formal definitions of aesthetics, to its success as a communicative act, where style contributes as much to meaning formation as the subject matter. However, prior image captioning datasets are not designed to empower state-of-the-art architectures to model potential human impressions or interpretations of images. To fill this gap, we design an annotation task heavily inspired by image analysis techniques in the Visual Arts to collect 1,440 image-caption pairs and 4,320 unique annotations exploring impact, pragmatic image description, impressions, and aesthetic design choices. We show that existing multimodal image captioning and conditional generation models struggle to simulate plausible human responses to images. However, this dataset significantly improves their ability to model impressions and aesthetic evaluations of images through fine-tuning and few-shot adaptation.

## 1 Introduction

> "*We never look at just one thing; we are always looking at the relation between things and ourselves.*"
> — **John Berger** (1972)

Images are rich objects for the semiotic study of *connotation*, as well as downstream objects for scientific study, including affect (Panda et al., 2018), framing (Uluçay and Melek, 2021; Christiansen, 2018; Powell et al., 2015) advertising and persuasion (Ye et al., 2019; Joo et al., 2014), with ramifications for marketing research (Oswald, 2012),

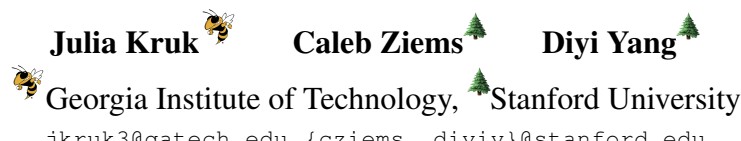

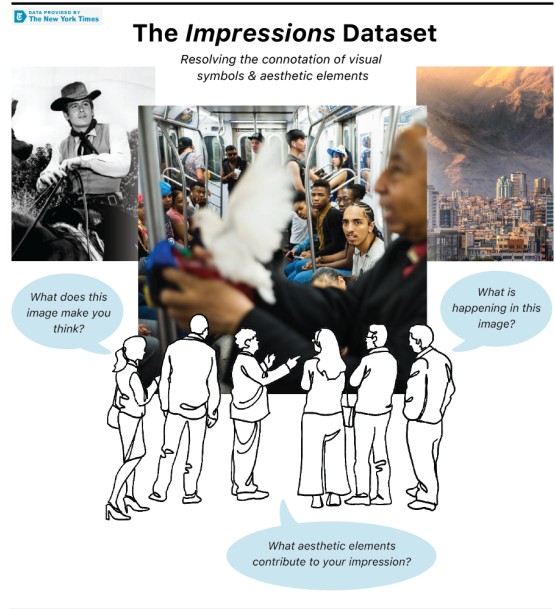

Figure 1: Photographs in **Impressions** contain a wide variety of styles and aesthetic elements. To understand the way these features contribute to visual semiotics and the perlocutionary force of the image, we gather human impression annotations inspired by image analysis techniques in visual arts and media studies.

public policy (Barnhurst and Quinn, 2012), journalism (Hullman and Diakopoulos, 2011), and communication more broadly. Current technological limitations have forced most studies to rely on manual analysis of connotation at smaller scales (see Hellmueller and Zhang, 2019 and others). While advances in image captioning have inspired Automatic Visual Content Analysis methods (Araujo et al., 2020), which excel at extracting objects, behaviors, and other *denotational* content, they often lack the awareness of how visual symbols *connote* non-literal meanings. This study presents a curated collection of image-caption data for training new systems that can understand connotation. In particular, we make explicit the image's *perlocutionary force*, or the meaning that is perceived by

---

[1] https://github.com/SALT-NLP/Impressions

an observer, as well as the aesthetic elements and concrete visual features that inspire these specific observer impressions (see Figure 1).

The image captioning datasets used to train multimodal architectures such as BLIP (Li et al., 2022), M2 Transformer (Cornia et al., 2020), and Flamingo (Alayrac et al., 2022), contain terse and reductive captions that describe their visual counterparts. This is because annotators were encouraged to write single-sentence captions that focused on the most concrete elements of an image: objects, entities, shapes, colors, etc. For tasks such as object detection and scene segmentation, this is the intuitive approach. But as transformer-based multimodal architectures grow more proficient, training solely on datasets such as COCO Captions (Chen et al., 2015), Conceptual Captions (Sharma et al., 2018), and WIT (Srinivasan et al., 2021a) will inhibit models' ability to reason about the semiotics of images. This motivates our **Impressions** dataset, which captions images with semiotic and aesthetic elements that ground viewers' subjective perceptions (see Figure 2)

As highlighted by Berger (1972), an observer cannot view an image without imposing upon it values, beliefs, and expectations that are highly grounded in their understanding of cultural norms and social cues. Humans instinctively make assumptions about emotions, actions, and relationships – setting, space, and circumstance – past, present, and future. Furthermore, these inferences are not solely dependent on the concrete objects and entities in an image. Photojournalists often manipulate composition, lighting, exposure, and camera angles to direct an audience's perception of the subject matter. Aesthetic design choices shape the meaning of an image, and can communicate just as much as the concrete elements depicted. To better understand this phenomenon, Barthes (1972) applied the principles of *denotation* and *connotation* to visual symbols in photography. *Denotation* is the meaning carried by the literal depictions in the image. Whereas *connotation* is the additional meaning assigned to a signifier that can be dependent on (1) rules and conventions the observer is familiar with, (2) visual and aesthetic elements of the image, or (3) cultural implication of the depiction as opposed to what is really there. Few datasets have ventured to encourage pragmatic inferences on visual scenes and collect audience interpretations and impressions of images. Yet an

understanding of an audience's likely interpretations of visual media could empower multimodal dialogue systems, where images can be used in combination with text to communicate. Furthermore, a better understanding of the connotation of visual symbols can enable text-to-image architectures to capture more nuance in language, thus rendering more salient compositions and styles in the output.

To this end, we contribute the following:

- The **Impressions** dataset, a multimodal benchmark that consists of 4,320 unique annotations over 1,440 image-caption pairs from the photography domain. Each annotation explores (1) the aesthetic impactfulness of a photograph, (2) image descriptions in which pragmatic inferences are welcome, (3) emotions/thoughts/beliefs that the photograph may inspire, and (4) the aesthetic elements that elicited the expressed impression.

- We show that state-of-the-art transformer-based architectures struggle to model human impressions and resolve aesthetic elements that sit at the core of the connotation.

- By leveraging the **Impressions** dataset, these architectures attain the ability through few-shot learning or fine-tuning. In a human evaluation task, the greatest improvements are on GIT (Wang et al., 2022), BLIP (Li et al., 2022), and OpenFlamingo (Alayrac et al., 2022), for which annotators preferred fine-tuned/adapted model generations over 80% of the time across all caption categories.

- We release an additional dataset of 50,000 image and leading paragraph pairs collected from the New York Times (NYT) official API for unsupervised exploration of visual semiotics and aesthetic element contribution to the coded iconic.

## 2 Related Work

**Visual Semiotics and Communication** This work is heavily inspired by the discussion of the semiotics of visual media in Barthes (1972), specifically how connotation and denotation of signifiers are extended from linguistics to visual studies. Additionally, in focusing on gathering human impressions we analyze the *perlocutionary force* of images (their meaning as perceived by an audience).

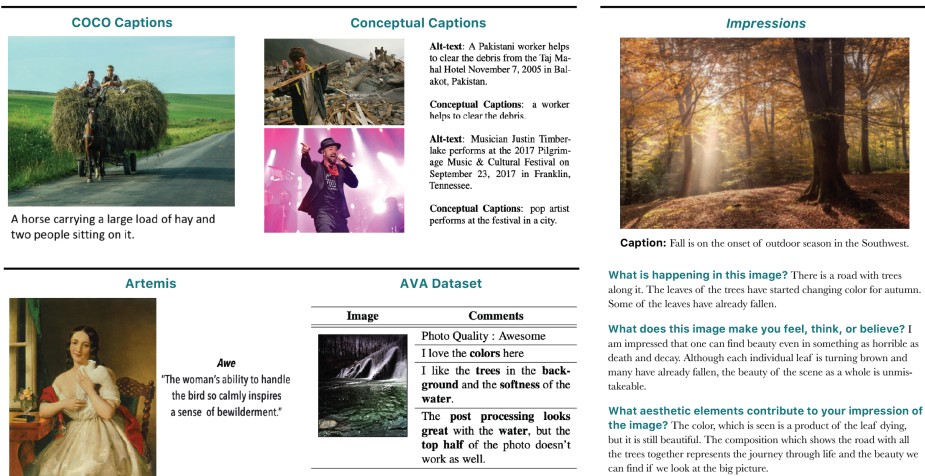

Figure 2: Comparison of **Impressions** dataset on the *right*, to COCO (Chen et al., 2015), Conceptual Captions (Sharma et al., 2018), Artemis (Achlioptas et al., 2021), and AVA (Murray et al., 2012) datasets in respective order on the *left*. Each visualization is taken directly from the cited works. In contrast to existing benchmarks, **Impressions** contains rich commentary on viewer perception that is not constrained to emotion, and a discussion of aesthetic elements that is grounded in the impressions that inspire.

Perlocution is among the types of speech acts, or communicative acts, identified by Austin (1962).

**Image Captioning**   Most image captioning tasks are framed with literal descriptions of the constituent objects, such as their behaviors, quantities and attributes. In the popular COCO (Chen et al., 2015), WIT (Srinivasan et al., 2021b), and Conceptual Captions (Sharma et al., 2018) benchmarks, these descriptions have a neutral, objective tone. Flickr30k (Young et al., 2014) takes it a step further, presenting crowd-sourced annotations of images depicting people going about everyday activities in an effort to study the visual denotation of linguistic expressions. More recent work *conditions* image caption generation on non-neutral stylistic features like humor (Chandrasekaran et al., 2017), romance (Tan et al., 2022; Gan et al., 2017), sentiment (Li et al., 2021), or personality traits (Shuster et al., 2019; Park et al., 2017).

In a similar effort to our own, Achlioptas et al. (2021) built a dataset for affective captions to media in the visual arts domain. The most important difference between this work and our own is that we consider additional impressions beyond affect. Not every image is going to give the viewer an emotive response but could alternatively inspire them to think or believe something. Most image captioning systems lack knowledge of symbolic iconography, cultural conventions, etc. (Lang and Ommer, 2018). By opening the question of human impression in this way, we aim to resolve a better understanding of social queues, cultural norms, and popular connotations of visual symbols through annotator responses. A breath of prior work has endeavored to generate pragmatic image captioning through context-agnostic supervision (Vedantam et al., 2017), listener and speaker models (Andreas and Klein, 2016), relaxation of language encoding (Zarrieß et al., 2021), and explicit image-caption pair annotation (Tsvilodub and Franke, 2023). Interestingly, datasets such as Flickr30k have been shown to unintentionally resolve annotator inference (Van Miltenburg, 2016), yielding more pragmatic and biased captions than intended.

**Aesthetic Image Analysis**   Some prior work has focused on aesthetic image captioning and analysis through collecting comments on aesthetic elements (Murray et al., 2012; Jin et al., 2019, 2022; Chang et al., 2017). These works approach style, design, and aesthetics from the perspective of beauty and visual appeal. The **Impressions** dataset was designed to ground the discussion of aesthetic elements in the impressions they inspire, to capture the link between the visual signifier and the signified. We also posit that aesthetic impact, or the likelihood of media to attract and hold attention, goes beyond what is considered classically beautiful and extends to the communicative utility of aesthetic elements. In other words, beauty and aesthetic impact are correlated but not the same.

Recent work has focused on empowering vision models to interpret and generate visual metaphors (Akula et al., 2023; Chakrabarty et al., 2023). Numerous datasets exist for art style (Khan et al., 2014; Florea et al., 2016), emotion (Mohammad and Kiritchenko, 2018), and aesthetic element classification (Amirshahi et al., 2015; Murray et al., 2012; Datta et al., 2006), as well as visual question answering with art (Garcia et al., 2020). There is active work on art classification and image description in the field of digital art history (Lang and Ommer, 2018). Although **Impressions** consists of photographic media, it also samples images from the creative field of photojournalism.

**Visual Question Answering.** Prior work has found great success in training and evaluating models' understanding of visual features and concepts through Visual Question Answering (VQA; Antol et al., 2015; Goyal et al., 2017; Das et al., 2017). Where early efforts leveraged questions focused on counting, object detection and concrete features, more recent works explore knowledge-based tasks (Marino et al., 2019; Zheng et al., 2021), pragmatic answer explanation (Zellers et al., 2019; Marasović et al., 2020), visual common sense (Krishna et al., 2017), and explainable social intelligence (Zadeh et al., 2019). Most notably, Liu et al. (2023) created a visual instruction tuning corpus with questions characterized by conversation, detailed description, and complex reasoning. The questions of our **Impressions** dataset closely resemble the complex reasoning category and could be used as an injection into such VQA corpora to improve impression generation and aesthetic impact reasoning.

## 3 Impressions Dataset

The **Impressions** dataset consists of 1,440 images and 4,320 distinct impression annotations. Each annotation is composed of three free-form responses to questions addressing (1) image description, (2) viewers' impression of the image, and (3) aesthetic image evaluation grounded in the viewers' impression. Additionally, we present aesthetic impact scores (in a range of 1 to 4) for 3,450 images. In the section below we will describe our process for collecting, annotating, and analyzing the dataset.

### 3.1 Collection

To build a rich resource for the semiotic study of aesthetic impact, we first need a set of impactful images whose styles and aesthetic elements vary.

Photojournalism is the use of photographs to both interpret a visual scene and to tell an impactful news story (Newton, 2013). Thus we anchor our collection around photojournalistic images from articles in the New York Times, a US media source with a longstanding reputation for quality (Teitz, 1999; Vise, 2011). **Impressions** contains 50,000 anchor images that we extracted from the publicly-accessible NYT API.

To introduce additional stylistic and artistic variation, we use the Google search API to retrieve 3 semantically-related images for each NYT anchor image. Our search queries come from perturbations of the original NYT image captions. We use three different perturbation methods: (1) one-sentence summaries produced via a BART conditional generation model (Lewis et al., 2020), (2) extracting key entities via the `spaCy 2` NER pipeline (Honnibal and Montani, 2017), and (3) constructing minimal dependency parse trees from anchor captions, defined by their subject, root verb, and any direct objects. Further information on caption perturbation methods can be found in Appendix A. After retrieving images for each perturbed query, we filter results, using only the top 3 images whose ViT (Dosovitskiy et al.) embedding similarity with the anchor image surpasses a pre-defined threshold.[2]

### 3.2 Annotation

The *aesthetic impact* of an image is its ability to draw and hold attention, as well as inspire the viewer to feel or think something. The first step of our annotation pipeline is designed to identify the most impactful images. Annotators consider a set of 4 semantically-related images (see §3.1) and rank them in the descending order of their aesthetic impact. Ties between two images are allowed, but to encourage more careful discrimination between images, we do not allow three and four-way ties. Images that have a mean aesthetic impact rank of 2 or greater are selected for free-form impression annotation. The variability of annotator rankings of aesthetic impact is characterized by an intraclass correlation coefficient (ICC1K) of 0.509, which demonstrates moderate annotator agreement.

The **Impressions** annotation process is specially chosen to scaffold the discussion for individuals with very little to no experience in media or visual arts as they consider the connotations of visual

---

[2] An image was kept if cosine similarity was less than 0.8 and greater than 0.2 with all other images in the set. Further discussion of these thresholds can be found in Appendix B.

signifiers and the *coded iconic* (Barthes, 1972), or portrayed story, of each image. As such, the design is heavily inspired by the fields of visual arts (Dondis, 1974), art history (d'Alleva, 2005), and historical image cataloging (Zinkham, 2006). Our specific prompts were:

1. **Image description**: 2-3 sentence response to the question *"What is happening in this image?"* Unlike the captions in prior works, these descriptions can contain information not explicitly depicted but rather inferred.

2. **Image Perception**: 2-3 sentence response to the question *"What does this image make you think, feel, or believe?"* In this way we aim to resolve the perlocutionary force of the image and its visual signifiers, in a manner not constrained to emotion alone.

3. **Image Aesthetic Evaluation**: 2-3 sentence response to the question *"What aesthetic elements contributed to your impression of the image?"* This way we ground the discussion of aesthetic elements in the audience's impression, drawing the connection between the signifier and signified.

Annotators were recruited from both Amazon Mechanical Turk and UpWork. The authors' motivations for leveraging these platforms is outlined in Appendix H. In total, we collect aesthetic impact rankings for 3,450 images, randomly sampled from our image collection process in §3.1. We collected free-form impressions for 1,440 of the most impactful images. A review of annotation instructions and examples can be found in Appendix G.

## 3.3 Quality Analysis

To highlight the distinguishing characteristics of the **Impressions** benchmark, we analyze sentiment, subjectivity, and concreteness of all impression annotations: the description, perception, and aesthetic evaluation. There does not yet exist a direct method for evaluating the richness and diversity of implied connotations in an image caption, but we expect that, compared to literal descriptive captions, connotation-rich image impressions will exhibit: (1) increased variance in the distributions of sentiment intensity, (2) increased subjectivity, and (3) lower concreteness scoring of linguistic data.

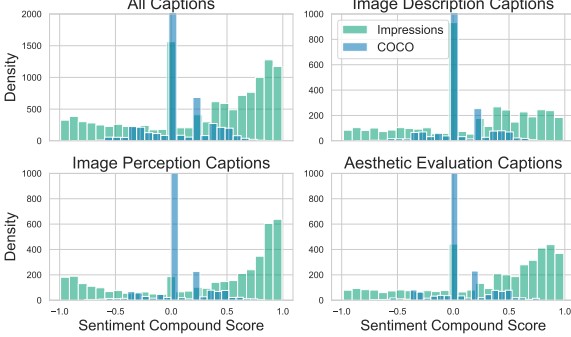

Figure 3: Evaluation of the sentiment intensity of captions in the **Impressions** dataset in reference to captions in COCO. The spread of Impressions caption sentiment is greater than that of COCO.

**Sentiment Intensity** Although we do not constrain impressions to emotion, we do expect that the distribution of sentiment intensity in impression annotations will exhibit a wider spread than what we see in traditional image-captioning datasets. We observe this in a direct comparison between Impressions and COCO captions in Figure 3. Notice that the distribution of our own Image Description annotations resembles that of COCO, which is expected. Whereas Image Perception and Aesthetic Evaluation annotations produce far more variable distributions of sentiment intensities.

**Subjectivity** A viewer's impression of an image is inherently shaped by the viewer's beliefs, and expectations, and sociocultural context, which inform their visual salience map and the connotations available to them. Thus we hypothesize that, compared with the literal descriptive captions, image impressions for a single image will be more variable. This variance is not meaningless noise, we posit, but rather meaningful sociocultural variation that is regularly bounded by the semantic frame of the image. We therefore expect that the variance of image impressions will be low, despite being greater than that of literal COCO captions.

In Figure 4, we see that the distributions for description, perception, and aesthetic evaluation captions in **Impressions** are wider than that of COCO. The median semantic variance for image descriptions (0.070), perceptions (0.088), and aesthetic evaluations (0.072) are all one order of magnitude greater than that of COCO captions (0.007). Still, given these small absolute values, we see that most image impressions are not highly variable.

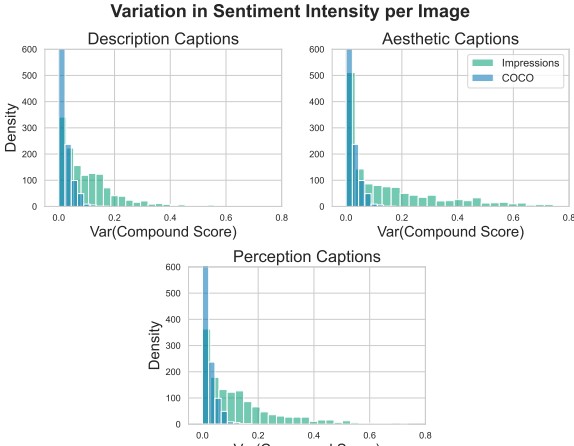

Figure 4: The variance in sentiment intensity of impression and aesthetic evaluation captions per image.

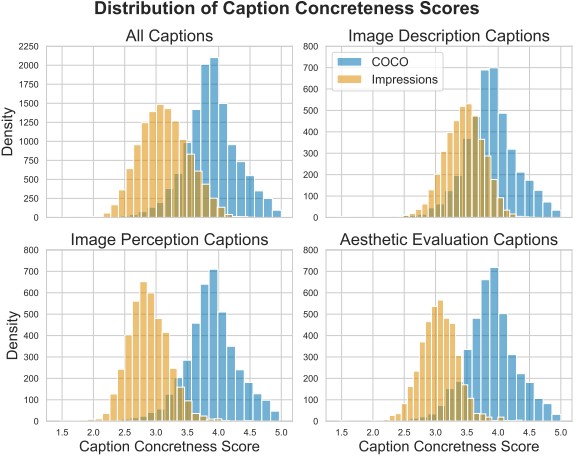

Figure 5: Distributions of caption concreteness scores in **Impressions** and COCO datasets.

**Concreteness**    The concreteness of a word is the degree to which the concept it signifies is an entity that can be perceived. We can define the concreteness of a sentence to be the average concreteness of its constituent tokens. Since image impressions are based on *connotations* which derive from the viewer's subjective relationship with symbols and non-literal referents in the image, we hypothesize that impressions will be more abstract, or less concrete, than traditional descriptive image captions.

To test this hypothesis, we compute token concreteness via the lexicon of Brysbaert et al. (2014), which contains concreteness scores for over 60k English lemmas, each on a scale from 1 (highly abstract) to 5 (highly concrete; based on experience). We find that each set of Impression captions (description, perception, aesthetic evaluation) is less concrete than COCO captions, each with a statistical significance of p < 0.001 by t-test. Visually, this is apparent in the gaps between Impression and COCO concreteness distributions in Figure 5.

## 4   Evaluation

To demonstrate the **Impressions** dataset's ability to enable state-of-the-art image-captioning and conditional-generation models to simulate human impressions and resolve impactful aesthetic elements of images, we design the following experiment. GIT (Wang et al., 2022), BLIP (Li et al., 2022), OpenFlamingo (Awadalla et al., 2023), and LLaVA (Liu et al., 2023) are fine-tuned / few-shot-adapted using a training set of 1,340 images and 4,020 impression annotations (each containing 3 unique captions). GIT and BLIP were fine-tuned

for image-captioning separately on each of the different **Impressions** data types (description, perception, aesthetic evaluation). OpenFlamingo was similarly few-shot adapted for VQA using 16 examples of one data type at a time. LLaVA was fine-tuned for VQA on all annotations simultaneously via an instruction-tuning training task.

**Automatic Evaluation**    Table 1 displays the BLEU, METEOR, and ROUGE scores for each of these models. Note that the **Impressions** dataset contains more variation in human-produced references than traditional VQA and image-captioning benchmarks. This quality makes CIDEr-D (Vedantam et al., 2015), a concensus-based evaluation of image descriptions, non-optimal for evaluating performance with this benchmark.

**Human Evaluation**    Each fine-tuned/few-shot adapted model is evaluated on 300 image-prompt pairs. The prompts are divided evenly between questions targeting image description, perception, and aesthetic evaluation. For the models fine-tuned for image-captioning, 100 captions are generated per annotation category. This same process is repeated on the base pretrained GIT, BLIP and LLaVA architectures, and in the zero-shot setting with OpenFlamingo. This produced a final set of 300 generation pairs, where each pair contains one caption from the fine-tuned or adapted model and one from the base. We submitted the 300 generation pairs to UpWork and requested that annotators identify which caption better simulates a potential human impression and identifies impactful aesthetic elements (see Appendix I). For each generation pair we collect 3 unique annotator votes

| | BLEU-1 | BLEU-2 | BLEU-3 | BLEU-4 | METEOR | R-1 | R-2 | R-L |
|---|---|---|---|---|---|---|---|---|
| GIT[†] | 0.272 | 0.156 | 0.090 | 0.054 | 0.165 | 0.295 | 0.106 | 0.246 |
| BLIP[†] | 0.347 | 0.195 | 0.111 | 0.068 | 0.184 | **0.325** | **0.111** | **0.261** |
| Open Flamingo (fs16)[*] | **0.500** | **0.275** | **0.159** | **0.098** | 0.231 | 0.318 | 0.093 | 0.246 |
| LLaVA-7b-v0[*] | 0.349 | 0.181 | 0.089 | 0.046 | **0.279** | 0.294 | 0.079 | 0.202 |

Table 1: Automatic token-overlap metrics for the four models fine-tuned on the **Impressions** dataset. These scores are computed on a scale between 0 and 1, over an evaluation set of 300 captions. [†] Denotes models fine-tuned for image-captioning. [*] Denotes models fine-tuned or few-shot adapted for conditional generation.

and select the majority vote as the final evaluation. The results of this human evaluation task are displayed in Table 2 in the form percentages at which annotators believed the model fine-tuned / adapted on **Impressions** produced a better caption.

## 5 Results

The results in Table 2 show that annotators preferred outputs from models fine-tuned/few-shot adapted with the Impressions dataset on average 76% of the time. The greatest improvements are observed with GIT, BLIP, and OpenFlamingo, for which annotators selected fine-tuned/adapted model generations over 80% of the time across all categories, but significantly more often for image perception and aesthetic evaluation generations. Marginal improvement is seen with LLaVA fine-tuned on the **Impressions** dataset, with annotators selecting generations from the fine-tuned model on average 56% of the time, most notably 59% on aesthetic evaluation generations.

**Image Description** Generations on image description were least improved by fine-tuning on the Impressions Dataset across all architectures explored in this study. However, as a plethora of datasets have been created for this purpose, this result was expected. The greatest improvement on descriptive generations was by fine-tuning GIT to achieve 75% preferential improvement over the base model. This was followed by fine-tuned BLIP, which was preferred 69% over BLIP base. This indicates that the Impression Dataset facilitated model learning of captions that most closely aligned with viewers' perspectives.

**Image Perception and Aesthetic Evaluation** The Impressions dataset helped improve GIT, BLIP, and OpenFlamingo on image impression and aesthetic evaluation generations. Since GIT and BLIP were pre-trained on corpora like ImageNet and COCO, their base performance was to generate more neutral, terse, and denotational captions that

were unable to convey human impressions or critical aesthetic elements. Although OpenFlamingo was competitive in producing image descriptions that aligned with human interpretations, it failed to follow instructions zero-shot when prompted on image perception and aesthetic elements. Provided 16 examples, the few-shot adapted OpenFlamingo was able to resolve human impressions reasonably well. Optimal performance was observed with 32 examples or more. BLIP and OpenFlamingo showcased the greatest improvement in aesthetic evaluation when fine-tuned/few-shot adapted on **Impressions**, producing preference scores of 88% and 100% respectively. GIT had the greatest improvements on image impressions, with a preference score of 92%. Qualitative comparisons of human impressions generated with different model architectures are displayed in Figure 6.

**LLaVA Performance** LLaVA was found to be incredibly competitive at reasoning about human impressions and aesthetic elements. Annotators expressed a marginal preference for generations from LLaVA-7b-v0 fine-tuned on **Impressions**, with the largest improvement observed in aesthetic evaluation (59% preference score). This architecture was pre-trained on a synthetic instruction-tuning multimodal dataset created using language-only GPT-4. Caption and bounding box information from the COCO dataset was leveraged in prompting GPT-4 to generate a conversation in which one neural speaker asks questions about the image, and the other answers. This dataset produced a model that excels at generating eloquent, figurative, and pragmatic descriptions of visual media. However, although LLaVA has made great strides in zero-shot VQA and overall language generation quality, we have found the model tends to miss connotation that is heavily grounded in style. It relies mostly on objects, entities, and setting to reason about potential human impressions. Yet there are instances where features such as contrast, lighting,

| | Description | Impression | Aesthetic Evaluation | All Captions |
|---|---|---|---|---|
| GIT[†] | 0.750 | 0.920 | 0.780 | 0.815 |
| BLIP[†] | 0.690 | 0.840 | 0.880 | 0.805 |
| OpenFlamingo-16[⋆] | 0.610 | 0.960 | 1.000 | 0.857 |
| LLaVA-7b-v0[⋆] | 0.560 | 0.530 | 0.590 | 0.560 |

Table 2: Results of the human evaluation task on base and fine-tuned/few-shot-adapted model generations. Each row shows the percentage at which annotators believed the model fine-tuned/few-shot-adapted on **Impressions** data generated better captions than the base/zero-shot model. Each column indicates the preference scores for individual caption types. [†] Denotes models fine-tuned for image-captioning. [⋆] Denotes models fine-tuned or few-shot adapted for the conditional generation.

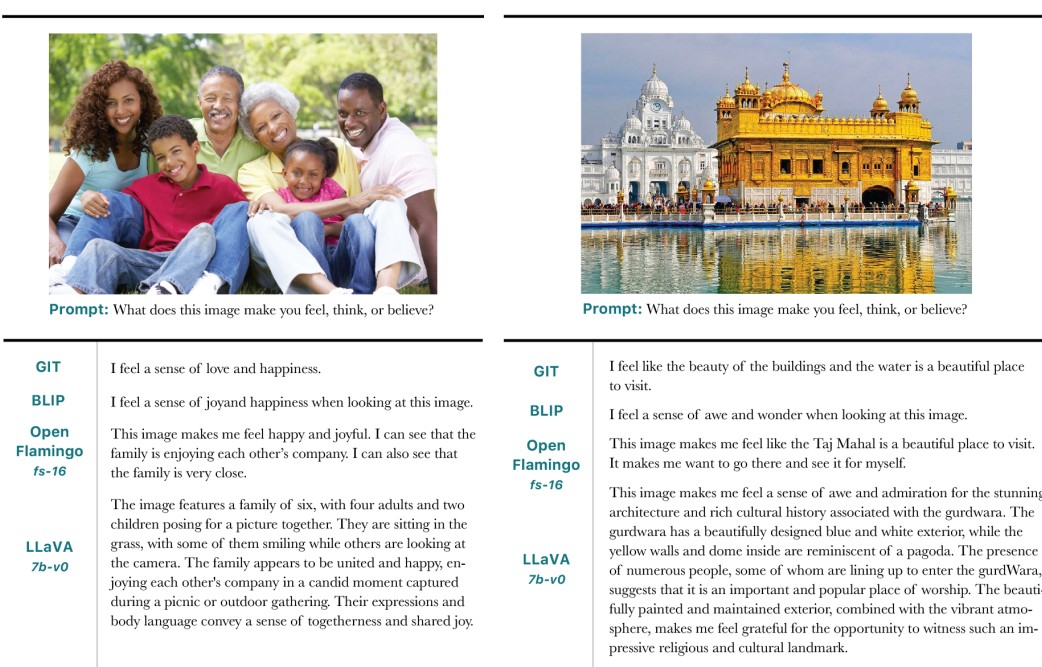

Figure 6: Comparison of each architecture's generation of potential human impressions on the **Impressions** dataset.

camera angle, and motion blur have great influence on perlocutionary force and the coded iconic. To address this weakness, **Impressions** could be injected into the complex reasoning category of this synthetic dataset. Additionally, we recommend that future exploration of synthetic visual instruction-tuning datasets leverages features pertaining to image aesthetics, in addition to bounding boxes, when prompting transformer-based dialogue systems. Such features could include pixel intensity (Datta et al., 2006), contrast (Ke et al., 2006), color distribution (Datta et al., 2006; Ke et al., 2006), or visual smoothness (Datta et al., 2006).

## 6 Persona-Specific Generation

To investigate the variation in human perceptions of images captured by **Impressions**, we design a set of experiments exploring the distinctive generation qualities that may emerge when training multi-modal models on annotations created by individuals belonging to different personality or demographic groups. Prior to beginning the image annotation task, annotators completed two surveys on personality traits and demographic information through Amazon Mechanical Turk (see Appendix E). To build each persona-specific LLaVA model of image perception, we fine-tune on a random sample of 500 annotations from the respective personality or demographic group: *introvert* vs. *extrovert*, *agreeable* vs. *disagreeable*, *business-oriented occupation* vs. *creative occupation*, and *no art experience* vs. *3+ years of art experience*.

An evaluation set of 100 images is leveraged to produce image perceptions with each of the eight LLaVA-7b-v0 models fine-tuned on persona-specific data. We then compare distributions of sen-

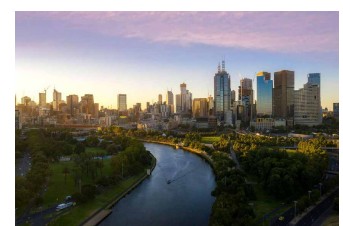

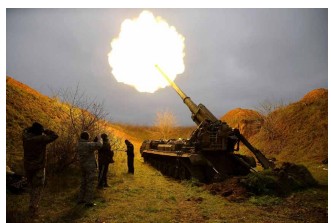

| | |
|---|---|
| **3+ Years Art Experience** | The image shows a cityscape with a line of trees along the river bank, a boat on the river, and high-rise buildings nearby. The blend of nature and urban environments creates a sense of harmony. It might evoke a feeling of relaxation and calmness in the viewer, as it portrays a peaceful coexistence between the city and the river. The viewer might also appreciate the architectural beauty of the high-rise buildings and the picturesque scene of the waterway with the boat on it. Overall, the image presents a visually appealing and harmonious combination of urban and natural elements. |
| **No Visual Art Experience** | The image showcases a city skyline with a body of water in the foreground, featuring boats and a small waterway. The sun is shining, casting warm light on the scene. This picturesque view makes me feel a sense of awe for the combination of urban architecture and natural elements. Overall, this captivating view evokes positive feelings and a desire to explore the city further. |

**Prompt:** What does this image make you feel, think, or believe?

| | |
|---|---|
| **Introvert** | The image shows a group of three men taking pictures of an army tank cannon that is being tested and shooting off fire. They are observing the explosion and capturing the moment with their cameras. The scene suggests that this might be a demonstration, possibly involving the military or a training exercise, where the tank's capabilities are being showcased. It must be fascinating and interesting to witness the power and effect of the cannon firsthand. |
| **Extrovert** | The image features a large military vehicle, possibly an army tank or artillery, spewing fire and smoke into the sky. This creates a dramatic display, reminiscent of a firework show. The smoke trails from the vehicle's exhaust give the scene a captivating visual effect. Several soldiers are standing around the fire-breathing military vehicle, watching the spectacle with interest. They seem to be fascinated by the impressive display, possibly wondering at the vehicle's capabilities or the purpose of the demonstration. Overall, the image conveys a sense of awe and excitement in the presence of the powerful military machine. |

**Prompt:** What does this image make you feel, think, or believe?

Figure 7: Image perceptions generated with LLaVA-7b-v0 fine-tuned with **Impressions** data created by annotators belonging to specific personality or demographic groups. The model fine-tuned on annotations created by individuals with 3+ years of visual art experience creates more descriptive image perceptions, with frequent reference to the viewer. The model fine-tuned on annotations created by annotators who identified as introverts produced more concise perceptions, whereas the extrovert model uses more metaphors.

timent intensity, sentence-level concreteness, and generation length to identify differences in model behavior across contrasting personas. We find that image perceptions created by the *extrovert* and *introvert* models have distinct distributions on generation length with a statistical significance of p = 0.026 by t-test. Similarly, the *no art experience* and *3+ years of art experience* models have distinct distributions on sentence-level concreteness scores with a statistical significance of p = 0.012 by t-test. Figure 8 illustrates the differences, with the extrovert model producing longer captions and the 3+ years of art experience model achieving slightly higher concreteness scores. A qualitative example of the differences in generated perceptions can be found in Figure 7. The remaining model pairs,

namely *agreeable* vs *disagreeable* and *business-oriented occupation* vs *creative occupation*, did not produce distinguishable distributions on any measure. It is important to note that more distinctive behaviors can arise as training and evaluation sets are scaled, although it is possible that certain personality or demographic traits do not correlate with unique image perception trends.

## 7 Conclusion

**Impressions** was designed with inspiration from visual arts and media studies to evaluate and enhance multimodal models' ability to reason about visual connotation. We show that, through fine-tuning and few-shot learning, this dataset enabled an array of architectures to resolve potential human impressions and discuss impactful aesthetic elements. State-of-the-art vision models are proficient enough at resolving entities and stylistic attributes to support such a task, but the weakness existed on the language side. This work highlights that targeted prompts modeled after image analysis techniques succeed in teasing out complex commentary on perlocation and the aesthetic elements it is grounded in. These prompts can be used by future works such as VQA and instruction-tuning, with applications in multimodal dialogue systems, text-to-image generation, and engagement prediction in advertising.

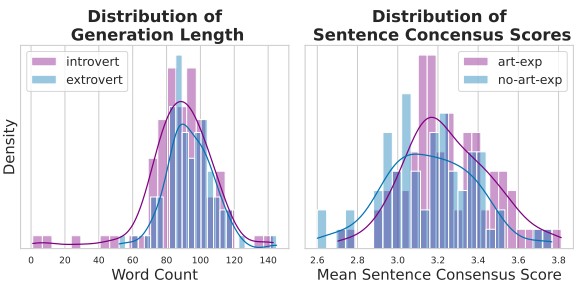

Figure 8: Distributions determined to be distinct with statistical significance p < 0.05 by t-test for model trained on contrasting personas.

## 8 Limitations

Perhaps the most noticeable limitation of the Impressions Dataset is its size. Due to resources constraints, the benchmark contains only 1,440 images with 3 unique annotations each. An increase in images would have allowed for a wider exploration of visual symbols, and an increase in annotations per image would have better resolved the natural variation in audience impression. Additionally, we acknowledge that by welcoming inference in the annotation task, we also risk introducing harmful, biased, and discriminatory ideas. Although the authors have not observed any such content in data quality checks, this dataset would benefit from an exploration on potential bias resolution.

## 9 Acknowledgments

We would like to thank all reviewers of this work for their insightful critiques, as well as the entire SALT lab for its ceaseless support. Special thanks to Ajay Divakaran for his mentorship, Benny Lin for his aid in building the annotation task UI, Brittney Newman for her insights as a domain expert in Photojournalism, Zac Crawford for his illustration on the front page, and all the annotators who contributed to the Impressions dataset. Caleb Ziems is supported by the NSF Graduate Research Fellowship under Grant No. DGE-2039655. This work was partially sponsored by NSF grant IIS-2247357 and IIS-2308994.

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

## A  Caption Perturbation for Semantically Similar Image Collection

To create image bins with marginal semantic similarity, but variation in curation, style, and design (or lack thereof), we collect 3 images via the Google Search API to accompany an anchor image from the NYT. The queries leveraged to search for these images are perturbed captions created from the leading paragraph associated with the NYT image anchor. Lead paragraphs are perturbed in the following three ways:

- **Summarization** Single sentence summaries of NYT anchor image captions are created via a BART conditional generation model (Lewis et al., 2020) that was fine-tuned on three different paraphrase tasks: Quora Question Pairs (Sharma et al., 2019), PAWS (Zhang et al., 2019), and the Microsoft Research Paraphrase Corpus (Dolan and Brockett, 2005).

- **Named Entity Recognition** Named entities were resolved from the anchor image caption by way of the `spaCy 2` NER pipeline (Honnibal and Montani, 2017). The method produced stand-alone entites as Google Image queries.

- **Subject Tree Extraction** We build queries by extracting from anchor captions their minimal dependency parse trees defined by their subject, root verb, and any direct objects contained in the caption.

A set of candidate queries is produced from these three perturbation strategies, from which three are selected at random. For each query, 5 images are collected via the Google search API. Finally, only one image is retained per perturbed caption. As discussed is §3.1 and Appendix B, images are vetted via thresholds on cosine similarity with other images in its intended image bin.

## B  Image Bins

The motivation behind creating imagine bins with semantic similarity but varying degrees of curation, was to group visual media that may be communicating similar things in different ways or to varying degrees of success. Reasoning about aesthetic impact in isolation is a difficult task even for the trained eye. But an annotator will find it easier to distinguish the saliency and communicative utility of visual media in a comparative setting.

We leverage both linguistic and visual information to collect images with semantic similarity. Images are collected from the Google search API via perturbed caption queries of the anchor image caption (see Appendix A). Additionally, an image is only added to a bin if has a cosine similarity less than 0.8 and greater then 0.2 to every other image in the bin. We experimentally determined that a cosine similarity larger than this range suggested the image was a duplicate, and anything lower was too dissimilar from the other visual media.

## C  Images of the Impressions Dataset

Figure 9 displays a random sample of images included in Impressions. As described in the paper, the dataset is a blend of photojournalistic images from articles in the New York Times attained through the official NYT AI, and images collected using the Google Search API. This data collection process yielded a wide variety of visual features, styles, and aesthetic elements.

## D  Annotator Qualification

Collecting clean and well written commentary on the semiotics of images demanded a complex annotation task. Utilizing crowd-sourcing platforms like Amazon Mechanical Turk provides access to a diverse collective intelligence for building machine learning resources. However, they also come with challenges in maintaining data quality (Zhang et al., 2022).

To ensure that annotators contributing to this dataset crafted well-writing and relevant commentary on perlocution and aesthetic elements, a qualification task was built through which their abilities could be reviewed before admission. It required 3 example annotations, in which the annotator had to rank images in a bin and then provide free-form annotations on the image they gave the highest impact score. Additionally, the authors conducted weekly quality checks on a random set of 50 submissions for every batch of 300 assignments.

## E  Annotator Personality and Demographics Data

As part of the data collection process, the annotators were asked to complete two additional qualifications through Amazon Mechanical Turk: a survey on demographic information and a condensed version of the Big 5 Personality test. These attributes were collected for the exploration of mod-

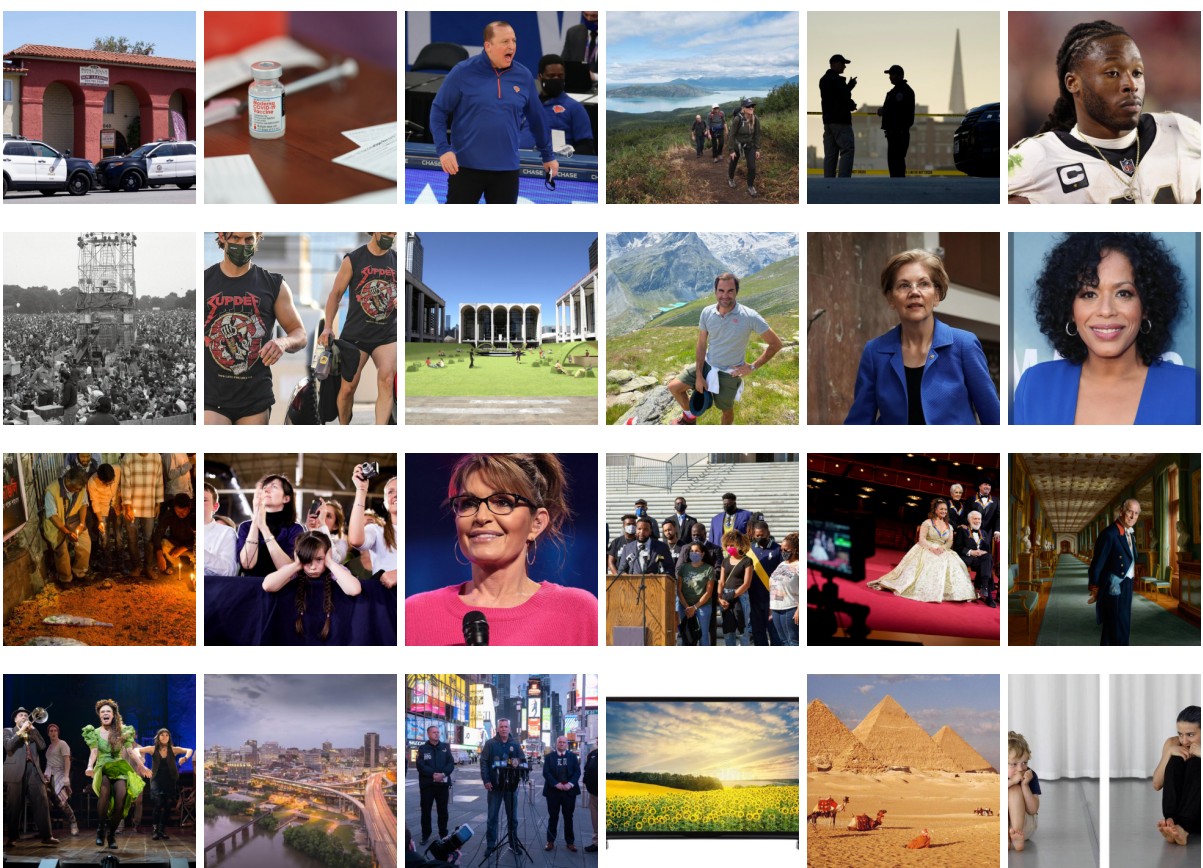

Figure 9: Images of **Impressions**. The following images were randomly sampled for the corpus.

eling viewer impressions of a particular community, and is included in **Impressions** metadata. Figures 10 and 11 showcase the personality traits and demographic attributes represented in the dataset.

## F Persona-Specific Generation Experiments

As discussed in Section 6, we investigate the distinctive behaviors that may arise when fine-tuning LLaVA-7b-v0 models on **Impressions** data created by annotators belonging to varying personality and demographic groups. Figures 12, 13 and 14 display distributions on sentiment intensity, sentence-level concreteness, and generation length for each model pair explored in this experiment. Note that although some distributions may appear different, the only ones that are distinct with statistical significant p < 0.05 are *introvert* vs *extrovert* on generation length, and *no art experience* vs *3+ years of art experience* on sentence-level concreteness.

## G Instructions for Annotation

Annotators were provided with the following instructions for the aesthetic image ranking task. Accompanying positive and negative annotation examples are shown in Figure 17.

In this task, we will be reviewing the set of 4 images displayed below and ranking them on aesthetic impact relative to one another.

A photograph is ***aesthetically impactful*** if the style and design choices catch your attention, and inspire you to feel, think, or believe something. Although subject matter is often important, we ask that you focus on how the visual elements (composition, lighting, color, perspective, etc.) impact your perception.

Rank each image on a scale of **1 to 4** based on how aesthetically impactful it is relative to the other images in the set. **1** would be the most aesthetically impactful of the set, and **4** would be the least impactful. If you strongly believe two photographs are equally aesthetically impactful, you may give them the same score. DO NOT give the same score to more than 2 images.

Instructions provided for the free-form an-

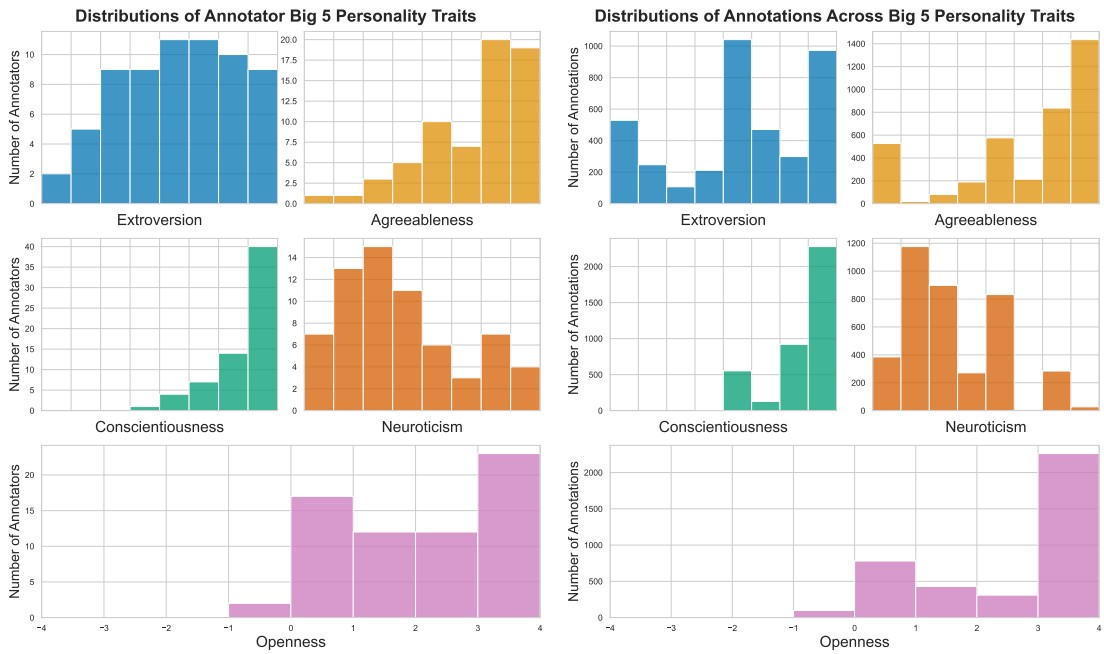

Figure 10: The figure on the *left* demonstrates the distributions of annotators over the Big 5 Personality Traits. Whereas the figure of the *right* showcases the number of image captions created by annotators characterized by specific Big5 Personality traits.

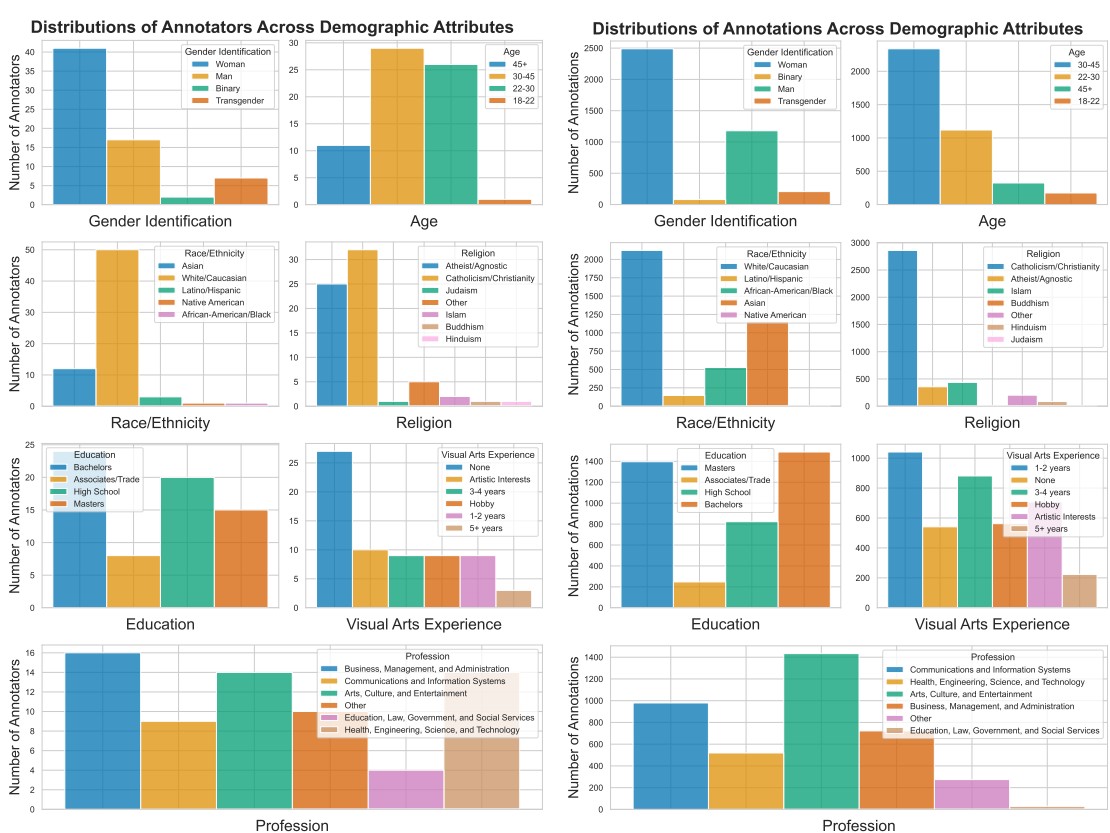

Figure 11: The figure on the *left* demonstrates the distributions of annotators over demographic categories. Whereas the figure of the *right* showcases the number of image captions created by annotators characterized by specific demographic categories.

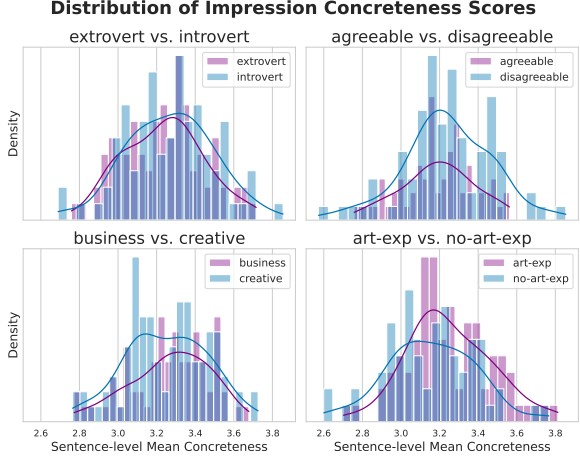

Figure 12: Sentiment intensity distributions on contrasting persona pairs.

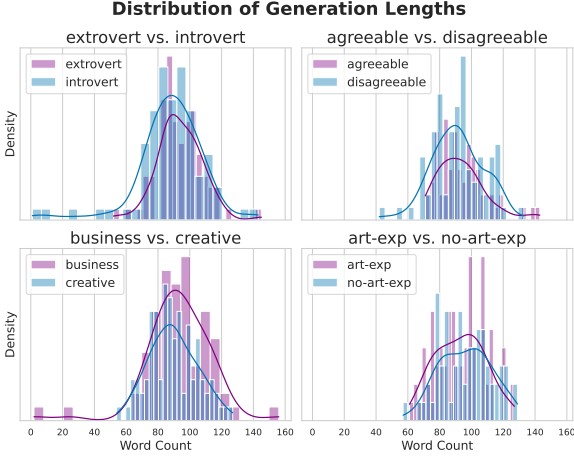

Figure 13: Sentence-level concreteness distributions on contrasting persona pairs.

Figure 14: Generation length distributions on contrasting persona pairs.

notation task on image description, image impressions, and aesthetic evaluation are shown below. Accompanying positive and negative annotation examples are found in Figure 16.

Given the image-caption pair below, please answer the following free-form questions. Your responses must be 2 - 3 sentences long. Review the positive and negative annotation examples below before beginning this task.

For the **first question**, please DO NOT simply paraphrase the caption. Remember the image can hold a lot of information too! Let us know what you understand about the scene after evaluating both the image and the text.

In the **second question** you will be asked about your impression of the image, your answer is expected to be subjective! Describe what this image makes you feel, think, or believe.

The **final question** will ask you to identify some visual elements in the image that contribute to your impression. We provide a list of common visual elements to assist you. You could use one, some, or none of the elements listed. It is up to you!

The list of visual elements provided is shown in Figure 15. This resource was created to assist annotators in answering the aesthetic evaluation prompt.

Figure 15: A collection of visual elements provided to annotators to guide them through linking concrete aesthetic features to their impressions of an image. Any visual elements outside of this list were welcomed.

## H  Crowd-sourcing Platform Commentary

Within the scope of this work, annotators were recruited via Amazon Mechanical Turk and Upwork.

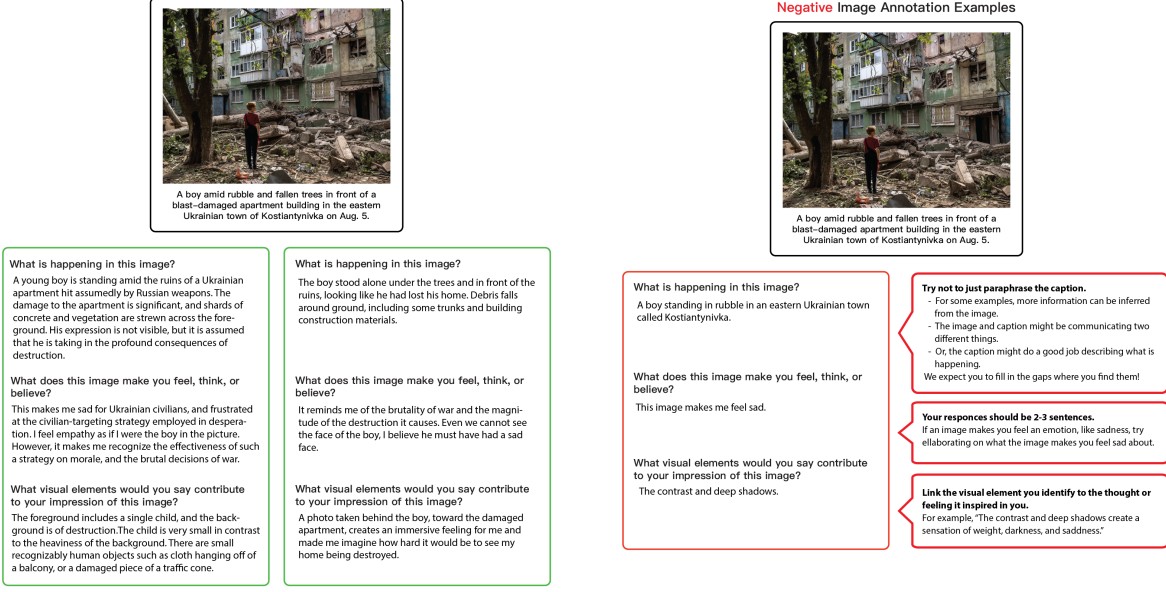

Figure 16: Positive and negative examples provided to annotators on aesthetic impact ranking.

The authors experienced the following important trade-offs of working with these different resources for attaining human annotation.

Amazon Mechanical Turk has a faster annotation turn-around time and is often less expensive as Workers are paid per completed assignment. However, quality control is a challenge as annotators are insentivized to complete assignments as quickly as possible, often to the detriment of annotation complexity and correctness. Even with qualifications in place and a manual evaluation procedure for admitting Workers, the authors experienced input quality decreasing overtime or completely changing after Workers passed the vetting procedures. Additionally, Mechanical Turk does not support effective avenues of communication with annotators. This makes it incredibly difficult to address annotator mistakes, misinterpretations, or abuse of the task.

After observing these behaviors on Amazon Mechanical Turk, the authors began recruiting contributors through UpWork in parallel. The onboarding process is considerably longer as one must review proposals, contract individual annotators, and provide instructions to each annotator individually. Hourly contracts prove to be more expensive, yet remove the incentive to complete assignments as fast as possible. This results in less data loss through quality vetting of completed annotations. If any mistakes or misinterpretations were uncovered during data quality reviews , these concerns could be addressed with annotators directly, which produced an increase in annotation quality over time. Additionally, once of team of over 8 annotators was consolidated on UpWork, the data annotation turn-around time caught up to that of Mechanical Turk.

In conclusion, although hourly contracts through UpWork were slightly more expensive and onboarding is more time consuming, the benefits to data quality and workflow made recruiting annotators through this platform the more efficient choice for data collection.

## H.1 Annotator Compensation

Assignments on Amazon Mechanical Turk were valued at $1.10, to amount to $15 per hour at the average annotation rate anticipated for this task (4 - 5 minutes). The annotation rate was estimated by recruiting a number of individuals (including the authors themselves) to complete the task while keeping time.

Annotators recruited through UpWork were paid hourly at a rate of $15 per hour. There were a couple of cases where annotators were contracted at $12 per hour. This was only arranged if the observed annotation speed was considerably lower than expected, but the annotator was eager to contribute to the task. Given that one annotation is expected to take less than 5 minutes, this avenue was only explored if an annotator is observed to take more than 10 minutes.

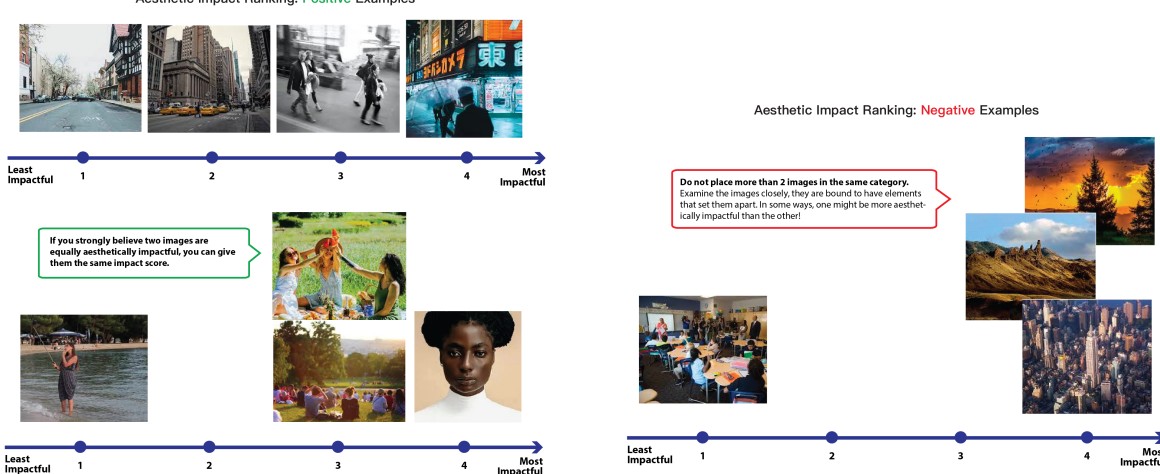

Figure 17: Positive and negative examples provided to annotators on aesthetic impact ranking.

# I Human Evaluation

The human evaluation task presented an image, prompt, and two generated captions to annotators recruited through Upwork, and requested they select the caption they believed to be the best fit for the image-prompt pair. One caption/answer to the image-prompt pair was generated by a base, pre-trained architecture, and the other was generated by the same model fine-tuned or few-shot adapted on Impressions. The following instructions were presented to annotators defining what characterizes the "better" caption:

In the final stage of this project will be a human evaluation task, where you will be comparing the outputs of two Generative AI models given an image and a prompt. You must select which caption does a better job of answering the prompt given the image shown. This project was designed to enable AI architectures to better predict plausible human impressions of images, resolve more socially-aware information, and better discuss what aesthetic elements can make an image impactful. Therefore, when we say select the "better caption", this is what we mean:

**The caption that better describes the events, actions, relationships and feelings communicated by the image.** This means going beyond listing off the items and entities depicted, and discussing the context of the scene and including information that an observer can reasonably infer. This also means capturing social and cultural information.

**The caption that better simulates a plausible human impression of the image.** Did the caption get the mood right? Is the impression or thought it simulates likely to be shared by a human audience? How much depth is there to its description?

**The caption that better identifies aesthetic elements and is capable of discussing style.** In other words can it discuss contrast, camera angle, light, etc. Bonus points if it correctly specifies how that design choice can inspire an audience.

**The caption is relevant to the prompt.** It must at least try to answer the question you will see acompanying the image.

Please do NOT base your decision on the following attributes: **caption length, punctuation, run-on or cut-off sentences, minor repetitiveness, complexity of grammar, and minor mistakes** (such as miscounting objects, misidentifying entities or people, and hallucinating an object that might make sense in the scene, but its not actually depicted).