# OpenReview forum: "Impressions: Visual Semiotics and Aesthetic Impact Understanding"
_EMNLP/2023/Conference — EMNLP 2023 Main_

### Official Review · Reviewer_GJAP · 2023-08-05

**Soundness:** 3

**Excitement:**

3: Ambivalent: It has merits (e.g., it reports state-of-the-art results, the idea is nice), but there are key weaknesses (e.g., it describes incremental work), and it can significantly benefit from another round of revision. However, I won't object to accepting it if my co-reviewers champion it.

**Paper Topic And Main Contributions:**

This paper investigates if aesthetic beauty in an image can result in effective communication. Current image captioning models do not correctly predict human responses to images.

**Reasons To Accept:**

They have evaluated on the impressions dataset that contains rich commentary on viewer perception.

They conclude that captioning after image analysis can help describe complex aesthetic elements.

**Reasons To Reject:**

Pg 1, right col, line no 041, how is this paper used in journalism and communication. Give details?

Pg 2, right col, line no 121, there is no reference for Impressions dataset?

Table 1 legend does not explain any trend. Which is the best?

Some discussion on adjective-noun pairs and sentiments in emotions is needed. See for example, ‘Text-Image Sentiment Analysis.’ 2023, From: CICLing 2018: 19th International Conference on Computational Linguistics and Intelligent Text Processing,

**Reproducibility:**

3: Could reproduce the results with some difficulty. The settings of parameters are underspecified or subjectively determined; the training/evaluation data are not widely available.

**Reviewer Confidence:**

3: Pretty sure, but there's a chance I missed something. Although I have a good feel for this area in general, I did not carefully check the paper's details, e.g., the math, experimental design, or novelty.

---

> ### Author Rebuttal · Authors · 2023-08-29
>
> Re: “reference for Impressions dataset,” on line 121, Impression dataset is a novel contribution presented in our paper, therefore, we did not include a reference here. Section 3 describes our unique process to collect viewer perceptions of visual media, as well as concrete visual elements that inspire these perceptions.
>
> Re: “journalism and communication,” the manner in which this work applies to these fields is more thoroughly explored on lines 208 - 228:
> “The Impressions dataset was designed to ground the discussion of aesthetic elements in the impressions they inspire, to capture the link between the visual signifier and the signified. We also posit that aesthetic impact, or the likelihood of media to attract and hold attention, goes beyond what is considered classically beautiful and extends to the communicative utility of aesthetic elements…”
>
> Re: “Table 1… trend,” we bold the best performing model for each metric. OpenFlamingo and BLIP are the strongest performers on standard ngram metrics (ROUGE & BLEU), whereas LLaVA-7b-v0 outperforms the other models on harmonic mean of precision and recall metrics when incorporating the stemming and synonym matching of METEOR. GIT is the worst performing model across all metrics.
>
> Re: “discussion on adjective-noun pairs”, although a valid analysis, we argue that it would be supplemental given the findings already presented on the distribution of sentence-level concreteness in section 3.3. Both of these methods demonstrate an increase in the use of figurative and expressive speech. We are happy to clarify this in the revised version of this paper.
>
> Re: “discussion on… sentiments in emotions is needed”, in section 3.3 we present (1) a detailed comparison of sentiment intensity between Impressions and COCO, and (2) an analysis of the variance in sentiment intensity per image, indicating greater variation in perceptions described by disparate viewers (see Figures 3 and 4).

---

### Official Review · Reviewer_7iNP · 2023-08-05

**Soundness:** 4

**Excitement:**

3: Ambivalent: It has merits (e.g., it reports state-of-the-art results, the idea is nice), but there are key weaknesses (e.g., it describes incremental work), and it can significantly benefit from another round of revision. However, I won't object to accepting it if my co-reviewers champion it.

**Paper Topic And Main Contributions:**

The paper introduces a novel dataset, Impressions, designed to explore the semiotics of images and investigate how visual features and design choices evoke specific emotions, thoughts, and beliefs. The dataset consists of 1,440 image-caption pairs and 4,320 unique annotations. In addition, the paper show that the state-of-art transformer-based architectures struggle to simulate human responses to images, but fine-tuning and few-shot adaptation significantly improve their ability to model impressions and aesthetic evaluations. The paper concludes that targeted prompts inspired by image analysis techniques are effective in eliciting complex commentary on perlocation and aesthetic elements.

**Reasons To Accept:**

The paper is easy to follow and presents its content in a clear and organized manner.

The authors have taken great care in providing a detailed annotation process and comprehensive dataset statistics, enhancing the paper's credibility and transparency.

The experiment results offer insights into both the dataset's effectiveness and the limitations of existing models. The empirical evidence shows  how fine-tuning and few-shot adaptation significantly enhance the models' capacity to mimic human impressions and aesthetic evaluations of images.

**Reasons To Reject:**

While the paper excels in many aspects, a more thorough statistical analysis compared with other relevant datasets is lacking. Incorporating a comparison table would be preferable, as it would provide a clearer understanding of how Impressions stands in relation to other existing datasets.

Lack of comparisons.

**Reproducibility:**

4: Could mostly reproduce the results, but there may be some variation because of sample variance or minor variations in their interpretation of the protocol or method.

**Reviewer Confidence:**

3: Pretty sure, but there's a chance I missed something. Although I have a good feel for this area in general, I did not carefully check the paper's details, e.g., the math, experimental design, or novelty.

---

> ### Author Rebuttal · Authors · 2023-08-29
>
> Re:  “a clearer understanding of how Impressions stands in relation to other datasets”, we have originally discussed this in section 3.3 where we highlight the characteristics that distinguish Impressions from COCO through quantitative analysis. Impressions has:
>
> - Greater variation of sentiment intensity across annotations (see Figure 3).
> - Greater variation in perceptions described by disparate viewers for the same image (see Figure 4).
> - More figurative speech to describe complex feelings, ideas and beliefs, as demonstrated by lower concreteness scores (p < 0.001; see line 410 and Figure 5).
> - Greater complexity and length of annotations (see Figure 2)
>
> We further demonstrate the utility of the Impressions dataset by human evaluation of image-captioning and VQA tasks, since fine-tuning and few-shot adaptation with Impressions leads to better impression generations (see Table 2).
>
> Re: “statistical analysis compared with other relevant datasets”, as discussed above, section 3.3 contains extensive comparison of Impressions to COCO. Another popular method of comparing image-captioning datasets is displaying CIDEr-D scores in the tabular form suggested by the reviewer. A strength of Impressions is it captures the variation in viewer perception of images, however, this quality makes consensus-based evaluation and quality metrics like CIDEr-D non-optimal for evaluating this dataset (as discussed on lines 434 - 440). That is why we omit any statistical analysis over such metrics.
>
> That being said, we would be happy to extend the quantitative analysis comparing Impressions with COCO to an additional two datasets in the camera-ready paper.

---

### Official Review · Reviewer_p7vQ · 2023-08-07

**Soundness:** 4

**Excitement:**

4: Strong: This paper deepens the understanding of some phenomenon or lowers the barriers to an existing research direction.

**Paper Topic And Main Contributions:**

The paper introduces a new benchmark, Impressions, consisting of 1440 image-caption pairs and 4320 annotations capturing various aesthetic elements and impactfulness. The paper shows that the SOTA models find it difficult to model human impressions and fine-tuning on the proposed benchmark improves their performance.

**Reasons To Accept:**

The proposed dataset with the rich annotations is very interesting and would be highly useful for the research community. The proposed prompts in section 3.2 (image description, perception and aesthetic evaluation) nicely capture key aesthetic elements. Extensive automatic and human evaluation, analysis has been performed on SOTA baselines. Overall, the paper is solid with well-thought and well-written annotation pipeline and quality analysis.

**Reasons To Reject:**

I do not find any major weaknesses with this work. It would be nice to add more example images to the appendix. I am curious, if the paper explored generating the images from the dataset? What would be some of the challenges in image generation that are specific to the aesthetic aspects? Citation to the below work on visual symbols/metaphors is missing.

Akula, Arjun R., Brendan Driscoll, Pradyumna Narayana, Soravit Changpinyo, Zhiwei Jia, Suyash Damle, Garima Pruthi et al. "Metaclue: Towards comprehensive visual metaphors research." In Proceedings of the IEEE/CVF Conference on Computer Vision and Pattern Recognition, pp. 23201-23211. 2023.


**Reproducibility:**

4: Could mostly reproduce the results, but there may be some variation because of sample variance or minor variations in their interpretation of the protocol or method.

**Reviewer Confidence:**

4: Quite sure. I tried to check the important points carefully. It's unlikely, though conceivable, that I missed something that should affect my ratings.

---

> ### Author Rebuttal · Authors · 2023-08-29
>
> Thank you for your thoughtful recommendations. We will add more example images to the appendix and include missing citations in our camera-ready paper.
>
> Re: question about image generation experiments - this insight could inspire a new project. We posit that an understanding of only certain aesthetic elements would transfer directly to image generation tasks. It seems that generative models have learned the relationship between lighting/color-scheme and particular moods, but composition, use of space, and connotation of specific visual elements are much greater challenges.

---

### Meta-Review · Area_Chair_xXXe · 2023-09-14

**Recommendation:** 5

**Metareview:**

Three reviewers provide feedback for this paper and their reviews were with consensus. The reviewers found the dataset interesting and useful. They found the extensive evaluations to be well done. They appreciated the human evaluation. The details provided in the paper, particularly about the annotation process was appreciated. There were very few concerns pointed out by the reviewers, and these were fairly minor. Given this consensus in soundness and excitement, I recommend accepting the paper.

---

### Decision · Program_Chairs · 2023-10-07

**Decision:**

Accept-Main

**Comment:**

Three reviewers provide feedback for this paper and their reviews were with consensus. The reviewers found the dataset interesting and useful. They found the extensive evaluations to be well done. They appreciated the human evaluation. The details provided in the paper, particularly about the annotation process was appreciated. There were very few concerns pointed out by the reviewers, and these were fairly minor. Given this consensus in soundness and excitement, I recommend accepting the paper.